# Stress and Coping Strategies of Online Nursing Practicum Courses for Taiwanese Nursing Students during the COVID-19 Pandemic: A Qualitative Study

**DOI:** 10.3390/healthcare11142053

**Published:** 2023-07-18

**Authors:** Hui-Man Huang, Yu-Wen Fang

**Affiliations:** Department of Nursing, College of Nursing, Tzu Chi University of Science and Technology, Hualien 970302, Taiwan; hhuiman@ems.tcust.edu.tw

**Keywords:** online nursing practicum, stress, coping strategies, nursing students, COVID-19 epidemic

## Abstract

Background: The coronavirus disease (COVID-19) pandemic has resulted in significant changes in nursing education. Maintaining social distance could slow down the spread of COVID-19, and it was necessary, but it significantly reduced students’ hands-on clinical practice experience in healthcare settings. Traditional classroom teaching in schools has transitioned to distance or online learning methods, which significantly reduced students’ hands-on clinical practice experience in healthcare settings. Although distance education had been implemented for a long time, there are many problems and challenges to be resolved. The experiences and needs of nursing students in remote clinical training urgently require further understanding. Purpose: To understand the stress and coping strategies of online nursing practicum courses for Taiwanese nursing students during the COVID-19 pandemic. Methods: A qualitative research approach with purposive sampling was supplemented by snowball sampling. Semi-structured interviews were conducted, and the data were collected following the eight-step process outlined by Waltz, Strickland, and Lenz (2010). The participants were 12 nursing students on a two-year nursing RN-to-BSN program at a university in Eastern Taiwan, consisting of 11 females and 1 male student. Findings: The stress and coping behaviors of nursing students consist of four main themes, each with three subthemes, including “urgent changes”, “the gaps between online courses and practical operations”, “mixed feelings of joy and anxiety” and “unexpected gains”. Conclusions: The pandemic has impacted nursing students’ learning and living. Engaging in online nursing practicum brought about significant stress; nevertheless, students employed various coping strategies to navigate through this challenging period. The findings of this study would also help nursing educators understand the learning gaps in clinical practicum among students.

## 1. Introduction

When the first case of coronavirus (SARS-CoV-2) appeared on 8 December 2019, a global pandemic was triggered that continued until 2021 [1]. The maintenance of social distance has emerged as a key strategy in preventing the spread of the disease. As a result, traditional classroom teaching in schools has transitioned to distance or online learning methods [2,3,4]. According to the survey reported by the United Nations Educational, Scientific and Cultural Organization (UNESCO) (2020) [5], to ensure education was uninterrupted, approximately 1.6 billion students worldwide switched to take classes at home following the outbreak of the pandemic. The nursing departments of various schools in Taiwan also encountered a new era [6]. To maintain the continuity of study, distance or online learning became the only way for teaching and learning to continue [7,8].

Distance education refers to the utilization of online courses that enable students to engage in educational activities even when they cannot physically attend school [9]. To ensure the timeliness and convenience of learning for students, teachers used synchronous and asynchronous teaching platforms such as Microsoft Teams, Google Meet, Cisco WebEx, etc. Although distance education had been implemented for a long time, it still had many problems and challenges to be resolved [8]. For example, with insufficient technology and equipment, it decreases opportunities for interaction between teachers and students, cause inadequate learning and weakening students’ learning abilities [5]. Meanwhile, students might experience problems such as learning fatigue, reduced self-efficacy in learning, and poor academic performance [10].

Maintaining social distance can slow down the spread of COVID-19, and it was necessary [11]; however, it significantly reduced students’ hands-on clinical practice experience in healthcare settings [7,12]. The potential issues and impacts of changing the clinical learning process of nursing students to online/distance modes have not yet been adequately addressed. The experiences and coping behavior of nursing students in remote clinical training urgently require further understanding [13,14]. This qualitative research method used in-depth interviews to understand the stress and coping strategies of nursing students who undertook online nursing practicum courses during the COVID-19 pandemic.

### Literature Review

The COVID-19 pandemic has significantly affected social, economic, cultural, and educational activities, causing isolation, lockdowns, and social distancing. It resulted in a quick shift from face-to-face teaching in schools to online courses [2,3]. Nursing education modes and strategies were significantly affected, particularly in clinical practical training courses for nursing students [15,16]. As a result, faculty and students found themselves unprepared, leading to sudden cancellations, postponements, or conversions of nursing practicum to online or distance learning formats [17]. Maintaining a balance between social distancing measures and learning became challenging for nursing students, making it difficult for them to develop their core competencies amidst the COVID-19 pandemic [18].

Distance education is characterized as a structured instructional approach that utilizes various technologies to facilitate regular and meaningful interactions between teachers and students, or among students themselves, when face-to-face instruction is not possible [16]. Online learning offers increased flexibility in teacher–learner interaction, and simplifies the scheduling of class time and space. It enables classrooms to accommodate a larger number of students, fosters independent learning, and cultivates students’ confidence and skills in using computers and the internet [14,16,19]. Distance education not only provides both synchronous and asynchronous learning opportunities, but also guides interactions between instructors and learners [20]. This innovative teaching approach could enhance students’ learning motivation and concentration. It makes it convenient for students to participate [21,22,23]. Despite the long-standing implementation of distance education, numerous challenges and issues persist, including software compatibility, proficiency in information technology, the availability of hardware equipment, and the impacts on students’ learning experiences [16]. Furthermore, additional challenges arise from a lack of distance education equipment, unfamiliarity with internet devices, difficulties in effectively monitoring students who cannot physically attend or participate online, a significant reduction in face-to-face interactions, and an increase in emotional stress due to a decrease in social activities [8].

Clinical practical training is a crucial challenge and milestone in the professional development of nursing students, and their professional performance relies heavily on their practical clinical experience, especially in the pandemic [24]. Students believe that the best place to learn practical clinical skills is in the clinical environment and on experimental wards [13]. However, the COVID-19 pandemic has posed varying degrees of challenges to nursing students’ clinical training courses [15,25]. These include students’ inability to participate in nursing work in healthcare settings, and them missing the study-to-work transition period [12]. Without practical learning, 75% of the students were concerned that they felt unable to cope with future clinical practice. Nursing students’ development of important clinical skills, problem-solving and diagnostic reasoning abilities, professional intimacy, and teamwork capabilities would be limited by the pandemic and distance learning [26]. The potential long-term impacts of online nursing clinical training during the COVID-19 pandemic must be investigated [27]. The experiences of nursing students would help nursing educators understand the potential impacts and learning gaps [15]. The aim of the study was to help us understand the stress and coping strategies of Taiwanese nursing students undertaking online nursing practicum courses during the COVID-19 pandemic.

## 2. Materials and Methods

A qualitative research method was employed, and purposive snowball sampling was used to recruit nursing students. The qualitative data analysis was based on eight major steps set out by Waltz, Strickland, and Lenz (2010) [28], including “Define the universe of content to be examined”, “identify the characteristics or concepts to be measured”, ”select the unit of analysis to be employed”, “develop a sampling plan”, “develop a scheme for categorizing the content and explicit coding and scoring instructions”, “pretest the categories and coding instructions”, “train coders and establish an acceptable level of reliability”, and “perform the analysis”.

### 2.1. Study Design

This study adopted a purposive sampling method and collected data through one-on-one interviews. Participants were recruited using a snowball sampling technique. We reached out to students through Line messages and invited them to participate. Additional students were gradually introduced. When the interview data had reached saturation the researcher stopped recruitment. The interviewers conducted interviews with nursing students and analyzed the interview content. During the interview process, a semi-structured interview guide was used (Table 1). The interview process lasted from April to May 2023 for each interviewee. The entire interview process was recorded, and the duration of the interview process was approximately 60–90 min for each participant.

### 2.2. Study Participants and Setting

The overall online nursing practicum took place from 2020 to 2021 in Taiwan. This study focuses on participants who were enrolled in a two-year RN-to-BSN program at a university in Eastern Taiwan. These participants had undergone online nursing practicum during their five-year associate degree in nursing (AND) program. A total of 12 nursing students, including 11 female students and 1 male student, were included in the study. 

To ensure research ethics concerning vulnerable populations and power dynamics between teachers and students in Taiwan, the inclusion criteria were: (1) students aged 20 or above; (2) experience in remote clinical nursing practicums; (3) providing consent to record the interview content; and (4) no students from classes taught by the researchers. The exclusion criteria were: (1) students under the age of 20; (2) unable to use fluent Chinese or Taiwanese; (3) those without experience in online clinical nursing practicum. The research setting was a convenient private space chosen by the participants. The participants were aged around 21.6 ± 1.77, and 10 students had been undertaking the online nursing practicum, while combined online and classroom teaching were undertaken by 2 students (See Table 2). The reasons for remote learning were study from home (70.6%), diagnosed isolation (5.9%), and quarantine after contact (23.5%). The department with the highest percentage of online practicum experience was the medical ward, with 28.6%, followed by the surgical ward with 20%. The percentage of participants who had a combination of online and in-person (labs) experience was 30.4%.

### 2.3. Data Analysis

This study followed the evaluation method proposed by Lincoln and Guba (1985) [29]. Credibility was established through maintaining transparency with the interviewees. The interviews were recorded, and detailed field notes were taken. Interview prompts were utilized to elicit comprehensive responses and further elaborate on the answers. To ensure confirmability, the audio-recorded interviews were transcribed by the first author, and the second author reviewed the transcribed data for accuracy by listening to the audio recordings. Transferability was ensured by the third author to verify that the data accurately represented the perspectives of the interviewees.

### 2.4. Ethical Considerations

Before the study was conducted, approval was obtained from Hualien Tzu Chi Hospital’s Institutional Review Board (IRB112-080-B). The study process was carried out in accordance with research ethics principles. Prior to the research, the research process was explained to nursing students, and informed consent forms were obtained from the participants. Subsequently, the interview schedules and locations were arranged.

## 3. Results

Assessing the stress and coping behaviors of nursing students during the online nursing practicum resulted in four main themes, each with three to four subthemes, including “urgent changes”, “the gap between online courses and practical operations”, “mixed feelings of joy and anxiety” and “unexpected gains”, as shown in Table 3.

### 3.1. Urgent Changes

The student council was prepared in advance for their clinical internship training. However, the clinical internship program at the hospital was abruptly interrupted or suspended because of the pandemic.

#### 3.1.1. Unable to Enter the Hospital Setting

Owing to the pandemic, all nursing students were required to suspend their clinical nursing internships in medical units and switch to distance learning. Clinical internships could only be completed at home or on campus.

“I was notified suddenly that I could not go for an internship. At the time, we were not aware of the government’s epidemic prevention measures, so we followed the instructor’s orders and stopped the internship abruptly” (P4); “At that time, the epidemic suddenly became very serious, and hospitals started admitting only patients with COVID-19. Perhaps due to concerns about student safety, both the hospital and the school temporarily decided to shift all clinical internships to online courses” (P9).

#### 3.1.2. Adopting Different Teaching Methods from the Past 

The sudden switch to online nursing clinical training had prompted instructors to utilize a variety of teaching strategies, such as videos, lectures, group discussions, simulated scenarios, student-produced videos, and videoconferencing exams, to achieve the goals of the clinical training curriculum.

“Online courses made it impossible to practice nursing care skills in person. As a result, my teacher asked us to create a scenario, write a script, and film a video demonstrating the suction technique ourselves. To successfully complete my nursing practice course, I had to rely on my family members to play the roles of patients or family members and to help with filming” (P6); “Our teacher shared videos about community care and provided interview practice videos for everyone. With only two weeks left, we were now able to return to school and used health education leaflets and toolkits to practice technical exercises” (P10).

#### 3.1.3. Problems with Information Technology and the Internet

The internet had become an important teaching tool for online nursing internship courses. However, problems such as computer hardware, software, networks, and operating proficiency had emerged consecutively, resulting in synchronization difficulties, disconnections, or operational errors, which in turn affected the progress of the course.

“I often encountered problems while using hardware devices during online classes. At that time, the computers at home were relatively old and if we had to use Google Meet and Word simultaneously, the computer would crash or become delayed. Some students spent a lot of time trying to figure out how to use software such as Google Meet for oral presentations or video sharing, which resulted in interrupted classes or delayed class endings” (P1); “In online courses, not only me but also other students had encountered network problems and disconnections. In the beginning, we spent a lot of time fixing the computers and the internet” (P2).

### 3.2. The Gap between Online Courses and Practical Operations

The students felt that even with different teaching activities, it was difficult to compensate for the benefits of actual patient contact, such as communication and adaptability training. They deeply felt the difference between online courses and physical care when continuing with their clinical practice.

#### 3.2.1. Online Learning Is Not Clinical Practice Nursing

Students felt that the inability to provide actual patient care and hands-on training highlights the limitations of online courses in replacing practical clinical experience. Learning impressions were mostly limited to simulated situations and imagination.

“Personally, I feel that online courses lack practicality and the learning experience is not as reliable. The case questions and care problems are mostly based on imagination. Online courses cannot replace the actual contact with patients” (P11); “By taking care of real patients, we can directly observe many symptoms and problems in patients and perhaps uncover other clinical clues. These were learning experiences that could not be replaced by online courses” (P1).

#### 3.2.2. Lack of Training in Communication, Interaction, and Adaptability

As they could not have actual contact with real patients, even though they practiced communication skills and discovered patient problems while proposing corresponding measures in simulated situations, it did not replace the experience of interacting with patients and healthcare teams in clinical settings.

“Psychiatric nursing care required more communication skills and experience, but we were unable to enter the medical unit. When I returned to the clinical setting, I might not know how to communicate with the patient or respond to their needs and preferences” (P5); “Online courses were vastly different from actual clinical practice. I could quickly learn and know how to react in a clinical environment by directly interacting with patients. The experience of studying in the ward was also helpful for license exams” (P12).

#### 3.2.3. Transition from Virtual Settings to Practical

The students expressed that the virtual world lacked the sense of urgency and pace of working in a real hospital setting. When they returned to clinical practice, it was difficult for them to quickly adjust to the pace, and they needed to constantly remind themselves to return to normal steps.

“Sitting in front of the computer without the tension of clinical practice was more like the general way of teaching in a school classroom, with a slower pace. I admitted that I had no pressure, and my learning attitude had become somewhat casual” (P8); “It took me about a week and a half to get used to it when I returned to practice in the physical ward for the next echelon. I was not used to returning to the clinic suddenly; I felt super unfamiliar, and suddenly, my head was not working. It took me several days to return to my normal pace” (P5).

### 3.3. Mixed Feelings of Joy and Anxiety

Owing to the cancellation of clinical internships during the pandemic, a gap had been created that could not be filled. The students were concerned that incomplete clinical training could affect their future nursing performance. However, in the absence of clinical internships, some students viewed online courses as breathing spaces.

#### 3.3.1. The Regret of Missing out on Clinical Internships

Students stated that practical nursing training through online courses was like a puzzle with missing pieces. Owing to the pandemic, canceled clinical internships could not be rescheduled, which was an insurmountable problem that may even affect future nursing education.

“There was no way to repeat the nursing internship. It was impossible to compensate for the insufficient part of the online internship, which seemed empty. Arrangements for nursing practice by schools and the Ministry of Education tend to be ‘flexible use’. So most of the internship time was spent in a hurry or inexplicably” (P2); “I heard other students complain that it must be miserable to do nursing work in the future. Everyone said, ‘I would be miserable in the future’” (P9).

#### 3.3.2. Different Life beyond the Camera

The online learning format provided students with more leisure time and opportunities for laziness during the COVID-19 pandemic. Shortcomings in software and hardware had become excuses for some students to temporarily turn off their cameras and avoid participating in class.

“As cameras were not turned on during online courses, some people became lazy and others were unsure of what to do, leading them to engage in other activities such as doing chores or sleeping” (P11); “During the four weeks of the internship, I lived exactly the same life every day. I felt bored. The advantage of being at home is that you do not have to rush every day” (P12).

#### 3.3.3. Worries about Nursing Work in the Future

The learning progress in online courses was difficult to control, and the gap between them and actual clinical care was huge, resulting in lower learning effectiveness than expected. Students believed that a lack of opportunities for practical operations would have a significant impact on their future clinical work, causing anxiety and worry.

“Sitting in front of the computer in class, I could not control the learning effect”.

“If you did not control yourself, you would not learn anything. But I would still have to do clinical nursing work in the future” (P2); “I was concerned that I might not know how to deal with patient problems, and I hoped that new nurses will not be overwhelmed by such situations. If I had actual internship experience, I would feel less uncertain and more prepared” (P9).

### 3.4. Unexpected Gains

Not all online nursing internship courses yielded negative results. The pandemic had provided students with the opportunity to re-examine their theoretical knowledge or adjust their learning pace for nursing.

#### 3.4.1. Creating a Simulated Environment

Students changed their approach to learning at home by using role-playing, practicing nursing skills with household items, and creating their own substitutes for real medical equipment. They created realistic situations at home to achieve the maximum benefit from practicing the clinical skill.

“The teacher used videos to demonstrate bathing skills for newborns. At home, we also utilized stuffed dolls or pillows as substitutes for Resusci Anne to practice the techniques. We also used our stuffed dolls or pillows at home as Ann to practice the techniques. During the technical examination, each student aimed at the video camera and performed baby bathing techniques with plush dolls or pillows” (P5); “During technical operation practice, we created IV sets, medicine tanks, and more. To simulate a realistic setting, I even used clay to create fake perineum and wounds, and then colored them to make them look as real as possible. I think that during the pandemic, every nursing student had to turn their home into a makeshift nursing laboratory” (P6).

#### 3.4.2. Enhance Knowledge Learned in the Past

Through computer-based online teaching, a larger proportion of content was focused on theoretical knowledge. For students, this was a good opportunity to review what they had learned in the past and identify areas for improvement.

“In online learning, it could be difficult to understand the teacher’s teaching style and content. However, in online internship courses, the focus was more on reviewing theoretical concepts. This could be beneficial as students could spend more time on understanding the theory, rather than just learning the technical skills” (P7).

#### 3.4.3. Reducing the Stress of Interpersonal Interaction

Some students felt pressure from interpersonal interactions during the clinical care process; however, the online learning format reduced the effort and stress involved in dealing with others, thereby alleviating difficulties in building interpersonal relationships. For some students, this was an unexpected benefit of online learning.

“I was worried that I would perform not so well in this practicum, because I did not like this subject. For practical internships, I had to contact individual cases face to face, senior nurses and instructors, all were the interpersonal pressures for me, so online internships also have benefits” (P5); “I had heard that during the internship in obstetrics nursing, male nursing students might be rejected by the cases or their family members, and did not be able to get in touch with every case or practical skills” (P1).

#### 3.4.4. Buffering the Tension in Clinical Practice Training

In Taiwan, clinical practicum courses for nursing students were mostly arranged according to a series of schedules. Owing to the pandemic, students did not have to be present in a high-stress ward atmosphere and could skip practicum departments that they were not good at or did not like, which helped them alleviate the pressure of the clinical practicum.

“I wasn’t interested in maternity care, so I was somewhat relieved when the course moved online. I’ve always been worried about not being able to adapt to clinical practice due to my weak academic foundation. I didn’t need to interact with individual cases, I just complete the reports and assignments assigned. It was so lucky to me. I would perform better online than in practical training” (P6).

## 4. Discussion

According to Metin Karaaslan et al. (2022) [19], the successful implementation of online nursing practicum courses relies on key factors such as students’ economic status, access to technical equipment and the internet, as well as the availability of diverse asynchronous learning methods. During the pandemic in Taiwan, the level of infectious disease prevention increased, and work and classes were moved to online platforms. Providing both synchronous and asynchronous courses could increase the success rate of distance learning, especially because asynchronous courses offered more flexibility for teachers and students to teach and learn without being limited by time and space [19]. However, there was a lack of standardization in training procedures during the COVID-19 pandemic [18]. 

Savitsky et al. (2020) [24] found that nursing students were susceptible to high levels of anxiety due to concerns about infection, transitioning to online learning, and uncertainty about the future. Maintaining a stable educational structure, providing high-quality distance education, and regularly encouraging and supporting students were key methods available for nursing schools to reduce student anxiety [30]. Online nursing practicum courses originated from the original teaching syllabus, content, and methods. They used synchronous online communication software, such as Google Meet, Cisco WebEx, or Microsoft Teams. Owing to the sudden switch to online teaching methods, many people were caught off guard, and some students experienced issues such as disconnections, network overload, and poor transmission during the online learning process, which resulted in interruptions or delays in the course. This was consistent with the study by Metin Karaaslan et al. (2022) [19], which stated that to ensure the success of distance learning, students needed to have adequate information hardware and software, including computers, internet connections, and a suitable learning environment. Providing adequate internet resources was a necessary condition for the successful completion of online courses [22]. 

Nursing students generally considered clinical environments and simulation laboratories to be the best places to learn clinical care skills [13]. All clinical practicum courses were abruptly changed to online nursing practicum courses, and students had to adapt to different modes of learning in a short period of time. The study by Rohmani et al. (2021) [10] showed that more than one-third of students preferred to practice nursing skills in the laboratory. Some students felt that sitting in front of a computer without the pressure and pace of clinical practice lacked the practical experience of providing care. They believed that online courses were very different from clinical practice, and were concerned about the difficulty of obtaining complete practical training. 

Shorey et al. (2022) [26] argued that online courses struggled to connect theoretical knowledge with a practical setting, which might result in the insufficient development of students’ clinical skills, problem-solving abilities, and diagnostic reasoning. Sinacori et al. (2021) [30] suggested that combining traditional face-to-face teaching with distance learning was more beneficial for knowledge acquisition than using either teaching method alone. Various technologies could be applied in nursing education to enhance students’ practical skills and develop their decision-making and problem-solving abilities [27]. The limited interaction with patients which would affect the development of communication skills [7,26]. However, students would avoid the potential risk of contracting an infection in the hospital.

In this study, students were concerned about their performance in nursing work; for example, they feared their problem-solving abilities, diagnostic reasoning, professional intimacy, and teamwork would not be fully developed in the future, which was same as was found in the study of Abbasi et al. (2020) [13] and Shorey et al. (2022) [26]. Clinical and laboratory environments were more conducive to acquiring clinical and technical skills, which is consistent with the results of this study. Sinacori et al. (2021) [30] took a different perspective, believing that online learning enhanced the acquisition of clinical skills more than face-to-face teaching.

Online courses offer increased flexibility in terms of class time, and can result in cost and energy savings [26]. In the present study, students expressed the belief that online courses also saved them time on transportation. Khen et al. (2008) [31] highlighted that online learning could lead to feelings of isolation and a lack of support among learners, but it also enhanced their learning experiences. However, the lack of comprehensive learning materials available for online learning could potentially undermine an individual’s self-discipline and compromise the existing learning environment. Furthermore, online courses have the potential to result in an inequitable distribution of learning resources, reduced teacher–student interaction, increased learning fatigue, and diminished self-efficacy [5,7,8]. The prolonged use of computer technology products can also lead to digital fatigue, which may adversely impact learning outcomes [26,32]. In this study, a similar situation was observed wherein not all students utilized synchronous videos during online classes. Students acknowledged that they tended to become more relaxed and less engaged when they were not visible on camera. The utilization of internet-based platforms and online examination methods also posed challenges in verifying the authenticity of exams or preventing students from cheating [22,26].

Online courses offered students distinct learning opportunities, setting them apart from traditional instructional methods. They offered the advantage of not only enhancing students’ knowledge review capacities, but also bridging any previous learning gaps; this result was similar to that of the research conducted by Nabolsi et al. (2021) [7] and Abbasi et al. (2020) [13]. In this study, students experienced a sense of accomplishment through the creation of their own health education videos and learning materials. Stress during the clinical training phase for nursing students emerged from various factors [33]. The students in this study felt relieved because they were unable to enter the hospital because of the pandemic control measures. Online courses did not require complex interpersonal interactions or physical contact with patients from different departments. For example, male students might experience embarrassment while dealing with pregnant women. Therefore, online courses avoided troubles in clinical practical settings. Thus, from the student’s point of view, the online nursing practicum not only helps to gain negative experience, but it could also represent a new order for the clinical training program. In the nursing student’s department in Taiwan, the ratio of male students to female students is unbalanced, indicating a scarcity of male students. Thus, in this study we included more female participants.

### Limitation

The severity of the epidemic, cultural perspectives, the implementation of epidemic prevention measures, etc., were the most influential factors that affect nursing students’ learning experiences and perceptions. Additionally, this study did not employ a randomized sample of participants; it also focused on the AND program, employed few participants, and there was a scarcity of male nursing students, which could lead to bias.

## 5. Conclusions

The stress and coping behaviors exhibited by nursing students during online nursing practicum can be categorized into four main themes, each consisting of three to four subthemes: “urgent changes”, “discrepancies between online courses and practical application”, “mixed emotions of joy and anxiety” and “unexpected gains”.

During the COVID-19 pandemic, participants had to make decisions regarding their careers or further education. Engaging in online nursing practicum brought about significant stress; nevertheless, students employed various coping strategies to navigate through this challenging period. Online learning facilitated the acquisition of professional nursing knowledge and improved learning outcomes for students. Additionally, it fostered independent learning, and alleviated the burden of pressure and difficulties encountered during clinical training. This study revealed that nursing students would create their own simulated learning scenarios at home, which may be specific to Taiwanese nursing education.

The participants were from different schools, with various instructors and teaching strategies for online courses, which may contribute to the saturation of the study results. Despite uncertainties surrounding the impact of nursing work on their future, the study’s findings suggest the standardization of online nursing practicum models and the incorporation of blended learning activities, such as flipped classrooms, gamification, and virtual reality, to meet the educational needs of nursing students. Online courses eliminate the need for complex interpersonal interactions or physical contact with patients in various departments. This study gathered valuable insights from students’ experiences in online nursing practicum, providing nursing educators with valuable inputs for the design of remote/online nursing courses. 

## Figures and Tables

**Table 1 healthcare-11-02053-t001:** Interview questions.

1. During the COVID-19 pandemic, what kind of impact on your nursing courses?
2. During the COVID-19 pandemic, what kind stress appearance in your clinical practicum course? What changes have been made in your clinical practicum course?
3. Has it been switched to distance, online, or some other strategies? How to arrange the 8 h (one-day) practicum scheduled in your nursing courses?
4. What are the teaching activities that you have received during the online nursing practicum?
5. During the online courses, do you have any problems with computers, the internet, or software that affects your learning?
6. What kind coping strategies have used for your online nursing practicum?
7. What do you think about the advantages and disadvantages of online nursing practicum?
8. After this practicum experience, what are your suggestions for an online nursing practicum?
9. What kind of activities is helpful for improving your nursing competency?
10. If the experience of online nursing practicum would impact your future nursing licensure ex-amination or nursing work? And how?

**Table 2 healthcare-11-02053-t002:** Background information of the participants. (n = 12).

Variable		SD/%
**Age**		21.6 ± 1.77 *
**Gender**	Female	10(83.3)
	Male	2 (16.7)
**Time of Practicum**	2021	8 (66.7)
	2022	4 (33.3)
**Department**		
Medical		10 (28.6)
Surgical		7 (20.0)
Gyn. and Obs. **		4 (11.4)
Pediatric		5 (14.3)
Psychiatry		4 (11.4)
Community Health		5 (14.3)
**Reasons for remote**		
Study from home		12 (70.6)
Diagnosed isolation		1 (5.9)
Quarantine for contact		4 (23.5)
**Practicum ways**		
Online		12 (52.2)
In-person (labs)		4 (17.4)
Both		7 (30.4)

* SD; ** Gyn. and Obs. means Gynecology and Obstetrics.

**Table 3 healthcare-11-02053-t003:** Themes and sub-themes of stress and coping strategies in nursing students.

Theme	Sub-Theme
Urgent changes	1. Unable to enter the hospital settings 2. Adopting different teaching methods from the past 3. Problems with information technology and the internet
The gap between online courses and practical operations	1. Online learning is not clinical practice nursing 2. Lack of training in communication, interaction, and adaptability 3. Transition from virtual settings to practical
Mixed feelings of joy and anxiety	1. The regret of missing practicum 2. Different life beyond the camera 3. Worries about nursing work in the future
Unexpected gains	1. Creating a simulated environment 2. Enhancing knowledge learned in the past 3. Reducing the stress of interpersonal interaction 4. Buffering the tension in clinical practice training

## Data Availability

Not applicable.

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
