# Peer review of "Stress and Coping Strategies of Online Nursing Practicum Courses for Taiwanese Nursing Students during the COVID-19 Pandemic: A Qualitative Study"

_healthcare, 2023, doi:10.3390/healthcare11142053_

Round 1

Reviewer 1 Report

The abstract should be written in accordance with the journal's guidelines. The study's background might be reduced and focused in order for the authors to provide a concise summary (including sample size, sampling method, dependent and independent variables included in the investigation) in the methodology. The current method of input was too superficial and difficult to understand. Certain parts of the result section must be paraphrased. The findings should back up the conclusion.

The content of the introduction should justify why the study is being conducted. This part is so long that forget the aim. Please write only one page and exactly write about the gap, main subject and aim of study. As a result, how can you say that Stress and coping strategies  can affect online nursing practicum courses? Please explain. Kindly use this articles 

COVID-19 Knowledge, Attitude, and Precautionary Practices among Health Professional Students in Oman.

Acceptance, attitudes, and barriers of vaccine booster dose among nursing students: A multicounty survey

Materials and Methods

lines 122-125 must inculded in study design. 

lines 128-130 must be in data collection not in study design. 

The qualitative data analysis was based on eight major steps by Waltz, Strickland, 127 and Lenz (2010) [28].  What is the eight steps and how used it in ur study. 

purposive snowball sampling was  used to recruit nursing students.  including 11 female students 148 and one male student. How you were used snowball sampling and majority of ur participants is female, also you did not consider about bias. Also you must mentioned that in Limitation. 

there are no data collection. 

explain how u invite the students. 

explain the duration of interview process, 

Is it online of face-to face. 

how many interviewees attend. 

is the interview were  individual or groups. 

Rigor and trustworthiness and Data analysis, is wrong.  kindly write again. kindly  read this  article and cited it will help you, for how to write;  Nursing Care and Barriers for Prevention of Venous Thromboembolism in Total Knee and Hip Arthroplasty Patients: A Qualitative Study. 

Author Response

Response to Reviewers

Thank you for the opportunity to re-submit our manuscript, titled “Stress and coping strategies of online nursing practicum courses for Taiwanese nursing students during the COVID-19 pandemic: A qualitative study”. We have carefully reviewed the reviewers’ comments and made revisions accordingly. Our response to reviewers’ comments is attached, with the revised portion highlighted in red. Please contact me if you have questions or need additional information. Thank you very much.

Reviewer 2 Report

Line 1-4. Title: Stress and coping strategies of online nursing practicum courses for Taiwanese nursing students during the COVID-19 pandemic: From nursing students’ perspective.

-        I suggest to authors to chance the title to: Stress and coping strategies of online nursing practicum courses for Taiwanese nursing students during the COVID-19 pandemic: A qualitative study.”

Line 146. Study Participants and Setting. Authors should be explaining the sample distribution (female vs male) and the response rate of the sample. Furthermore is not understandable to the reader the phrase “The participants in this study were 12 nursing students in a two-year nursing course in an RN-to-BSN program at a university in Eastern Taiwan, including 11 female students and one male student. The online nursing practicum period was from 2020 to 2021. At that moment, these nursing students were in the final stage of a five-year associate degree in nursing (AND) program”, please describe further the duration of course.

In addition, could be added by the authors the way of approached the participants? (by personal contact, e-mail). And before the start of each interview, if participants were oriented about the aim of the study, confidentiality, and data management practices and if the all participants gave their consent to participate in the study.

Line 168 results: suggested to the authors to add a table with the demographics of the participants.

Line 336-443: Authors should discuss the results of the present study and how they can be interpreted in perspective of previous studies and of the working hypotheses. The findings of the present study and their implications should be discussed in the broadest context possible and limitations of the work highlighted. Future research directions may also be mentioned by the authors (such a psychological condition of the students after the pandemic and the impact of e-learning).

Suggested to authors to summarize in one part the section of discussion without subparagraphs.

Enhance the discussion with the follow references:

 1. doi: 10.1007/s11482-022-10095-3,

 2. DOI: 10.1186/s12909-020-02312-0

3. https://doi.org/10.1016/j.heliyon.2022.e11927

Line: 438 -443. Limitations: suggested to the authors to announced more limitations of the present study.   For example, the size of the sample of the participants, and generally the limits of the qualitative study e.g., a not randomized sample of the participants.

Line 475 Acknowledgments: suggested to the authors to include to the acknowledgments the students for their participation in this study.

References: all references must be reform, please follow the journal instructions (e.g. reference 29).

Author Response

(The authors gave the same response as above.)

Reviewer 3 Report

Thank you for the opportunity to review this manuscript. Very interesting and topical issue. I am impressed with the approach you have taken about reporting students concerns. 

I have a few comments:

Throughout the manuscript you include the date with the reference i text this is not required. 

Line 157 you mention external criteria and state mental instability participants were excluded how did you measure this and for those who demonstrated instability what did you do to support these students?  

Line 160 please explain further evaluation method re credibility etc how do you use this criteria in your study?

Line 298 what is Ann?

Line 343 sentence does not make sense.

Good luck 

Author Response

(The authors gave the same response as above.)

Reviewer 4 Report

This is a qualitative study on the perception of nursing students about the online practical course. The purpose of the study is clear, as well as the methods used. The data analysis method and sample size definition could be described in more detail. The results are described accordingly. The discussion could be improved by avoiding the repetition of results.

Some sentences could be improved, I suggest a comprehensive review in English.

Author Response

(The authors gave the same response as above.)

Reviewer 5 Report

Congratulations to the authors for investigating such an important topic for the nursing profession, since practices in health centers are fundamental.

The bibliography seems to me to be correct and up to date.

The choice of the sample by the "snowball" system should be better explained, since only 12 subjects participated.

One exclusion criterion is age, specifically under 20 years of age, but are there nursing students under 20 years of age? If so, it would be biased not to include them since they can also provide information. One could even analyze whether there are differences in responses according to age.

Do they describe the informed consent of the participants?

The conclusions are scarce. They should provide answers to the objectives and be clearer.

Partial repetition is 14.26%. Ideally it should be below 10%. Review the introduction and discussion.

Author Response

(The authors gave the same response as above.)

Round 2

Reviewer 1 Report

NONE

Reviewer 2 Report

Thank you for your corections.